# Internal Mammary Arteries as a Model to Demonstrate Restoration of the Impaired Vasodilation in Hypertension, Using Liposomal Delivery of the CYP1B1 Inhibitor, 2,3′,4,5′-Tetramethoxystilbene

**DOI:** 10.3390/pharmaceutics14102046

**Published:** 2022-09-26

**Authors:** Azziza Zaabalawi, Lewis Renshall, Frances Beards, Adam P. Lightfoot, Hans Degens, Yvonne Alexander, Ragheb Hasan, Haris Bilal, Brigitte A. Graf, Lynda K. Harris, May Azzawi

**Affiliations:** 1Department of Life Sciences, Manchester Metropolitan University, Manchester M1 5GD, UK; 2Division of Pharmacy and Optometry, University of Manchester, Manchester M13 9PL, UK; 3Maternal & Fetal Health Research Centre, University of Manchester, Manchester M13 9WL, UK; 4Manchester Academic Health Science Centre, Central Manchester University Hospitals NHS Foundation Trust, Manchester M13 9WL, UK; 5School of Pharmacy and Biomolecular Sciences, Liverpool John Moores University, Liverpool L3 3AP, UK; 6Institute of Sport Science and Innovations, Lithuanian Sports University, 44221 Kaunas, Lithuania; 7Department of Cardiothoracic Surgery, Manchester Foundation Trust, Manchester M13 9WL, UK; 8Faculty of Health and Education, Manchester Metropolitan University, Manchester M15 6BG, UK

**Keywords:** 2,3′,4,5′-tetramethoxystilbene, endothelial cell, reactive oxygen species, liposomes, oxidative stress, coronary artery, internal mammary artery, vascular function, coronary artery bypass graft, nanoparticle drug delivery, cardiac rehabilitation

## Abstract

A significant number of patients with severe cardiovascular disease, undergoing coronary artery bypass grafting (CABG), present with hypertension. While internal mammary arteries (IMAs) may be a better alternative to vein grafts, their impaired vasodilator function affects their patency. Our objectives were to (1) determine if inhibition of the cytochrome P450 enzyme CYP1B1, using liposome-encapsulated 2,3′,4,5′-tetramethoxystilbene (TMS), can potentiate vasodilation of IMAs from CABG patients, and (2) assess mechanisms involved using coronary arteries from normal rats, in an ex vivo model of hypertension. PEGylated liposomes were synthesized and loaded with TMS (mean diameter 141 ± 0.9 nm). Liposomal delivery of TMS improved its bioavailability Compared to TMS solution (0.129 ± 0.02 ng/mL vs. 0.086 ± 0.01 ng/mL at 4 h; *p* < 0.05). TMS-loaded liposomes alleviated attenuated endothelial-dependent acetylcholine (ACh)-induced dilation in diseased IMAs (@ACh 10^−4^ M: 56.9 ± 5.1%; *n* = 8 vs. 12.7 ± 7.8%; *n* = 6; *p* < 0.01) for TMS-loaded liposomes vs. blank liposomes, respectively. The alleviation in dilation may be due to the potent inhibition of CYP1B1 by TMS, and subsequent reduction in reactive oxygen species (ROS) moieties and stimulation of nitric oxide synthesis. In isolated rat coronary arteries exposed to a hypertensive environment, TMS-loaded liposomes potentiated nitric oxide and endothelium-derived hyperpolarization pathways via AMPK. Our findings are promising for the future development of TMS-loaded liposomes as a promising therapeutic strategy to enhance TMS bioavailability and potentiate vasodilator function in hypertension, with relevance for early and long-term treatment of CABG patients, via the sustained and localized TMS release within IMAs.

## 1. Introduction

Coronary artery disease (CAD) remains one of the leading causes of disability and mortality worldwide, accounting for an estimated 17.8 million deaths annually [1], with 30–70% of patients often presenting with hypertension [2]. One early marker of hypertension is endothelial dysfunction (ED), characterized by elevated oxidative stress and blunted vasodilation [3,4]. Recently, human and animal studies have demonstrated a substantial role for the cytochrome P450 enzyme, CYP1B1, in precipitating this pathology via increased generation of the potent vasoconstrictor 20-HETE and subsequent production of NADPH oxidase derived reactive oxygen species (ROS) [5,6,7,8]. Intervention by coronary artery bypass graft (CABG) surgery is used as a last resort for patients with severe CAD, with the endothelial function of bypass grafts being crucial to long-term graft patency [9]. While human internal mammary arteries (IMAs) have been shown to be a better alternative to vein grafts [10,11], impaired endothelial function may affect their patency, and coupled with increased vasoconstriction and vascular remodeling, can lead to graft failure [10,12]. Nonetheless, even when given specific vasodilators, these patients remain at risk of developing ischemic episodes. The intensive management of hypertension is one of the major elements of secondary preventative therapies that is crucial in the perioperative period to optimize graft patency and to maximize recovery after CABG [13].

Recent studies have explored novel derivatives of the polyphenol antioxidant, resveratrol (RV), as attractive supplements to current antihypertensive treatment strategies, with enhanced pharmacological activity and bioavailability [14]. One such derivative is the methylated analogue 2,3′,4,5′-tetramethoxystilbene (TMS), shown to be 1000 times more potent in inhibiting CYP1B1 associated with hypertension, reducing NADPH oxidase activity, ROS generation, and subsequently rescuing endothelial dysfunction in spontaneously hypertensive rats than RV [5,7,8]. Furthermore, there are numerous reports showing the potential of TMS as a therapeutic agent [14], without obvious adverse effects in normal cells [15], and without exhibiting cytotoxicity to human cells [16]. In addition, the development of nanotechnology has been a game-changer as a platform to deliver drugs to diseased tissue. PEGylated liposomes are one such approach, and have a prolonged circulation time compared to non-PEGylated nanomaterials [17]. Using isolated rat aortic vessels, we previously demonstrated that attenuated endothelium-dependent dilation following acute tension elevation was associated with increased ROS production. Attenuated dilation was successfully alleviated following incubation with TMS-loaded PEGylated liposomes via the reduction in NADPH oxidase derived ROS and potentiation of NO [18]. We also demonstrated sustained dilation (4 h after exposure to oxidative stress associated with elevated tension) compared to TMS solution [18]. The present study’s objectives were to (1) assess the potential of TMS, delivered via PEGylated liposomes, to restore dilation of IMAs harvested from patients undergoing CABG surgery, and (2) assess mechanisms underlying the mode of action of TMS, using coronary arteries from normal Wistar rats, using a physiologically relevant ex vivo model of hypertension. Our findings have important implications in the potential use of encapsulated TMS as a therapeutic intervention to potentiate vasodilator capacity and improve cardiac rehabilitation via sustained and localized TMS release within IMAs, following CABG surgery.

## 2. Materials and Methods

### 2.1. Synthesis of TMS-Loaded Liposomes

PEGylated liposomes were composed of 1,2-distearoyl-sn-glycero-3-phosphocholine (DPSC, 65 μmol/L), cholesterol (30 μmol/L) and 1,2-distearoyl-sn-glycero-3-phosphoethanolamine-N-[methoxy(polyethylene glycol)-2000] ammonium salt (DSPE-PEG, 5 μmol/L), which were dissolved in methanol: chloroform (9:1 ratio, Avanti Polar Lipids, Birmingham, AL, USA). Solvent was removed by rotary evaporation to produce a thin lipid film, which was rehydrated with TMS in sterile PBS (2 mL, 2 mM, Sigma Aldrich, Gillingham, UK). The suspension was heated to 55 °C for 4 h and vortexed hourly to produce large multilamellar vesicles. The suspension was extruded 15 times using a 1 mL Mini-Extruder (Avanti Polar Lipids, Birmingham, AL, USA) through a 0.2 μm, 19 mm polycarbonate membrane, surrounded by two 10 mm filter supports in a unilamellar liposome suspension. Unencapsulated TMS was removed from the formulation by dialysis against PBS (8 × 1 L, 24 h: Slide-A-Lyzer Dialysis Cassettes, MWCO 3.5 kDa). The formulations were stored at 4 °C until use. The average size (hydrodynamic diameter) distribution (SD) and polydispersity index (PDI) were measured by dynamic light scattering and calculated from three independent liposomal preparations (DLS, Zetasizer Nano ZS, Malvern Instruments Ltd., Malvern, UK). PDI is a measure of the degree of uniformity of the size distribution of the liposome suspension.

Our previous study showed promising effects of TMS-loaded liposomes in rat aortic vessels [18] while King et al. demonstrated efficient encapsulation of IGF-II by ELISA, suggesting 50% encapsulation efficiency [19]; hence, we used the same methodology to encapsulate TMS for the present study to demonstrate significant differences between functional effects of TMS-loaded liposomes and empty liposomes.

### 2.2. HPLC Analysis of TMS/TMS-Loaded Liposomes Stability

Following incubation of vessels with TMS-loaded liposomes or TMS solution, a storage buffer comprising NaH_2_PO_4_ (0.4 M at pH 3.6), ascorbic acid (20% *w*/*v*) and EDTA (0.1% *w*/*v*), was added (10% *v*/*v*) to aliquots of TMS-loaded liposomes or TMS solution. Extraction buffer (5 mL, NaCl (4% *w*/*v*), ascorbic acid (2% *w*/*v*), EDTA (0.001% *w*/*v*)) was combined with an equal volume of sample (TMS-loaded liposomes or TMS solution), then ethanol (10 mL) was added, and the tube was vortexed. Liquid extraction was performed by adding 25 mL hexane/BHT (250 ppm), then the solution was vortexed for 1 min, and the extraction was repeated. The hexane layers were combined and dried, redissolved in ethanol and analyzed by HPLC. All samples were extracted in triplicate.

HPLC analysis was conducted on a Shimadzu HPLC system, comprising a Prominence delivery system (with LC-20AB binary pump, SIL-20ACHT autosampler set to 8 °C, and CTO-20A column oven set to 30 °C) connected to a NexeraX2 (SPD-M30A) diode array detector, scanning between 190–700 nm. A 10 µL sample extract aliquot was injected onto a Kinetex 5 µm C18 100 × 4.6 mm column (Phenomenex, Macclesfield, UK) fitted with a matching precolumn. Samples were separated at a flow rate of 0.3 mL/min using a gradient consisting of MPA (5% acetonitrile 95% water and 0.2% formic acid) and MPB (95% acetonitrile, 5% water and 0, 2% formic acid), starting with a linear increase from 0–70% MPB from 0 to 15 min, 70–95% MPB from 15–25 min, 95–0% MPB from 25–26 min, and a recalibration at 0% MPB from 26–30 min. TMS (Sigma Aldrich, SMB00388, ≥98% purity) was quantified by a linear 7-point standard curve plot ranging 0.75–25 ng/mL at 325 nm. Limit of quantification (LOQ) and limit of detection (LOD) for TMS was below 0.1 ng/mL.

### 2.3. Physiological Function Studies

#### 2.3.1. Human Internal Mammary Artery (IMA) Functional Studies

Experiments were performed on the left internal mammary artery (IMA) harvested from patients aged 55–75 years undergoing coronary artery bypass graft (CABG) surgery within 1 h of tissue harvest. All subjects gave their informed consent for inclusion before they participated in the study. The study was conducted in accordance with the Declaration of Helsinki, and the protocol was approved by HRA and Health and Care Research Wales (HCRW) (IRAS approval 255023). Peri-vascular adipose tissue was carefully removed, and IMA segments divided into 3-mm rings then mounted between two fine steel wires in an organ-bath containing oxygenated physiological saline solution (PSS) at 35 °C, as previously described [18,20]. The contraction dose-response relationship was initially assessed by exposing vascular rings to increasing concentrations of phenylephrine (Phe, 10^−9^–10^−3^ M; Appendix A). In subsequent experiments, increasing concentrations of acetylcholine (ACh, 10^−12^–10^−3^ M), followed by sodium nitroprusside (SNP, 10^−12^–10^−3^ M), were applied to sub-maximally constricted vessels (Phe, 10^−5^ M), eliciting endothelium-dependent and -independent relaxant responses, respectively. In order to establish the minimal dose of TMS required to achieve maximal dilation, cumulative doses of TMS (1 pM–10 μM) were added to vessels pre-constricted with Phe (Appendix A). To establish the influence of TMS-loaded liposomes (1 nM) on vessel function, ACh and SNP responses were examined following 1 h incubation. To determine the dilator component potentiated by TMS-loaded liposomes, inhibition studies were performed using a pharmacological approach with the dilator pathway inhibitors; nitric oxide synthase inhibitor Nω-nitro-*L*-arginine (L-NNA, 100 µM), small and intermediate calcium-activated potassium channel blockers Apamin (100 nM) and TRAM-34 (1 µM) and cyclooxygenase (COX) inhibitor Indomethacin (10 μM), as illustrated in Appendix A. The mechanistic contributions of AMPK and SIRT-1 were also investigated following co-incubation with the inhibitors Dorsomorphin (20 μM) and EX-527 (10 μM), respectively.

#### 2.3.2. Rat Coronary Artery Functional Studies

First-order septal coronary arteries were excised from male Wistar rats, euthanized by stunning and cervical dislocation in accordance with the Animals (Scientific Procedures) Act 1986, and after Institutional Ethical Approval (SE161741). Arterial segments were mounted between two glass cannulas on a pressure myograph chamber (Living Systems Instrumentation, St Albans City, VT, USA), as described previously [21]. Arterial segments were mounted between two glass cannulas on a pressure myograph chamber (Living Systems, USA) and pressurized to an intravascular pressure of 60 mmHg using a servo-control unit (Living Systems Instrumentation, USA). All arteries were sub-maximally pre-constricted in serotonin (5-HT, 10^−6^ M; Appendix A). Endothelial-dependent (ACh) responses were assessed prior to and following acute pressure elevation (150 mmHg) for 1 h, in the presence/absence of 1 nM TMS-loaded liposomes, TMS solution, or blank liposomes. All controls were subjected to standard pressure alone. Endothelial-independent (SNP) responses were also examined. In a separate set of experiments, the sustained dilator response, 4 h after return from pressure elevation, was assessed. To evaluate the possible contribution of ROS to the attenuated dilation, vessels were co-incubated with superoxide dismutase (SOD, 300 U mL^−1^) or apocynin (NADPH oxidase inhibitor; 30 µM).

Inhibition studies were performed to determine the contribution of the dilator component following TMS-loaded liposome exposure, as well as their cellular uptake mechanism. Responses were assessed following vessel co-incubation with the dilator pathway inhibitors described in Section 2.3.1, as well as the endocytic uptake inhibitor chlorpromazine hydrochloride (25 μM), to assess the mode of uptake of TMS-loaded liposomes. The mechanistic contributions of AMPK and SIRT-1 were also investigated. To ascertain whether TMS-loaded liposomes act via endothelial-independent mechanisms, the endothelium was denuded via the introduction of 3–4 air bubbles prior to beginning the experiments and ACh responses reassessed.

### 2.4. Detection of Reactive Oxygen Species in Human Internal Mammary Arteries (IMAs)

The levels of cytosolic superoxide, mitochondrial ROS and endothelial NO bioavailability were quantified in human IMAs, in the presence/absence of TMS-loaded liposomes, using dihydroethidium (DHE; 10 µM) and mitochondrial superoxide indicator (MitoSOX; 15 µM) and DAF-FM (10 µM), respectively. Isolated IMAs were subsequently incubated in the presence/absence of blank-or TMS-loaded liposomes for 1 h. Control vessels were incubated in PSS alone. To enhance vessel permeabilization, IMAs were initially incubated with 1% Triton followed by 30 min incubation with the ROS/NO detection assays. Fluorescence was measured using a BioTek Synergy HT microplate reader.

For superoxide tissue detection using DHE and following 1 h incubation in either PSS, blank-or TMS-loaded liposomes, isolated human IMAs were embedded by orienting into Tissue-Tek OCT Compound, in a sagittal plane to expose the lumen, then frozen on dry ice and sectioned (8 μm) using a cryostat. Sections were incubated in the dark with 5 µM DHE staining solution for 30 min. After washing slides with deionized H_2_O, coverslips were mounted using ProLong™ Diamond Antifade Mountant with DAPI. Sections were imaged at 10× magnification using Leica Thunder Microscope at excitation/emission wavelengths of 520/610 nm. The Integrated Thunder Analysis System was used to quantify fluorescence signal intensity (mean intensity/area).

### 2.5. Immunohistological Analysis of Diseased Human Internal Mammary Arteries (IMAs)

Isolated IMAs were fixed with 4% paraformaldehyde, embedded in paraffin, and sectioned (8 μm) using a microtome. Elastic Van Gieson’s (EVG) staining was carried out following standard protocols. For alpha smooth muscle actin (α-sma) and the endothelial cell marker Ulex Europaeus Agglutinin I (UEA I) immunofluorescent staining, sections were deparaffinized, and rehydrated. Antigen retrieval was performed by incubation in 0.01 M citrate buffer (pH 6) at 60 °C for 30 min. Tissues were then blocked with 1% bovine serum albumin in phosphate-buffered saline (PBS) for 1 h and incubated with Alexa Fluor 488 conjugated primary antibodies anti-α-sma and UEA I conjugated with Rhodamine. Slides were washed in PBS and mounted with ProLong™ Diamond Antifade Mountant with DAPI. Images were captured using Leica Thunder Microscope.

### 2.6. Statistical Analysis

Statistical analysis was performed using Prism version 9 for Windows, GraphPad Software, La Jolla, CA, USA. All values are presented as mean ± standard error. Data were tested for normal distribution using Shapiro–Wilk test. Results were analyzed using a two-way ANOVA followed by Tukey’s multiple comparisons post-test. Statistical significance was * *p* < 0.05, ** *p* < 0.01, *** *p* < 0.001, **** *p* < 0.0001.

## 3. Results

### 3.1. Synthesis and Characterisation of TMS-Loaded Liposomes

Blank liposomes had a mean diameter of 148 ± 1.3 nm and a polydispersity index (PDI) value of 0.117 in PBS. TMS-loaded liposomes had a mean diameter of 141 ± 0.9 nm and a PDI value of 0.113. All measurements were conducted at pH 7.2–7.4. In TMS-loaded liposomes, TMS did not significantly degrade after 4 h (TMS concentration 0.131 ng/mL at 0 h (± 0.018) vs. 0.129 ng/mL at 4 h (± 0.016)), while in solution, TMS concentration was reduced by 20% after 4 h (0.109 ng/mL at 0 h (± 0.004) vs. 0.086 ng/mL at 4 h (± 0.013); *p* < 0.05) (Appendix A).

### 3.2. TMS-Loaded Liposomes Potentiate Endothelial-Dependent Dilator Capacity of Isolated IMAs from CABG Patients Ex Vivo

There was no significant difference in Phe-induced pre-constrictor responses between control IMAs and those incubated with TMS-loaded liposomes (Appendix A). Isolated IMAs showed only 20% dilation at the highest ACh concentration. Incubation with TMS-loaded liposomes significantly improved ACh-induced vasodilation compared to PSS and blank liposomes (Figure 1A), while endothelial-independent responses to SNP were not significantly affected (Appendix A). Co-incubation with L-NNA (Figure 1B), Indomethacin (Figure 1C), Apamin and Tram (Figure 1D), all four inhibitors in combination (Appendix A), Dorsomorphin (Figure 1E) or EX-527 (Figure 1F) abolished the dilator responses potentiated by TMS-loaded liposomes.

### 3.3. TMS-Loaded Liposomes Potentiate Endothelial-Dependent Dilation via Activation of NO and Endothelial-Dependent Hyperpolarization (EDH) Pathways, via AMPK in Isolated Rat Coronary Arteries Ex Vivo

All vessels constricted in response to high potassium salt solution (KPSS) and serotonin (sub-maximal; 5-HT 10^−6^ M). Acute pressure elevation significantly reduced endothelial-dependent responses, likely attributable to increased ROS production, since alleviation of dilation inhibition was confirmed following co-incubation with SOD or apocynin (Figure 2A). The reduced dilation following acute pressure elevation was also alleviated after incubation with TMS solution or TMS-loaded liposomes, while blank liposomes had no significant effect (Figure 2B). Furthermore, the effects of TMS were maintained for 4 h after exposure to TMS-loaded liposomes, but not to TMS solution alone (Figure 2C). Endothelial-independent dilation was not significantly affected (Figure 2D).

Co-incubation of TMS-loaded liposomes with LNNA, or Apamin and Tram attenuated dilation (Figure 3A,B, respectively), while incubation with Indomethacin had no significant effect on ACh-induced responses (Figure 3C). Co-incubation of TMS-loaded liposomes with both LNNA and Apamin and Tram (Figure 3D), or all four inhibitors in combination (Appendix A) abolished dilator responses. Endothelial denudation completely abolished ACh-induced responses (Figure 3E).

Co-incubation with Dorsomorphin attenuated dilation (Figure 4A) while EX-527 had no significant effect (Figure 4B). Co-incubation with Chlorpromazine hydrochloride in the presence of TMS-loaded liposomes partially reduced dilator responses (Figure 4C).

### 3.4. Treatment with TMS-Loaded Liposomes Reduces ROS Moieties, Restoring NO Bioavailability in Isolated Human Internal Mammary Arteries (IMAs) Ex Vivo

Following co-incubation of IMAs with TMS-loaded liposomes, levels of superoxide anions and mitochondrial ROS were reduced, reflected by the significant reduction in fluorescence (Figure 5A,B, respectively), with an increase in the relative fluorescence of intracellular NO (Figure 5C), compared to untreated IMAs, or exposure to blank liposomes. Vascular superoxide levels were determined in situ in human IMA sections, using the redox-sensitive dye, DHE. Incubation with TMS-loaded liposomes resulted in a significant reduction in DHE fluorescence, indicating attenuated superoxide generation, when compared to the stronger signal observed in IMAs exposed to PSS alone or blank liposomes (Figure 5D,E).

EVG staining suggested arterial remodeling, which could be attributed to increased deposition of matrix proteins in the vessel wall. Whether long term exposure to TMS-loaded liposomes could delay the remodeling process, however, has yet to be confirmed. Immunofluorescence labelling with UEA I illustrated the absence of an intact endothelial lining (Figure 6B; white arrow), suggesting endothelial damage.

## 4. Discussion

Our key finding is that liposomal delivery of TMS alleviates the ROS-mediated-attenuation of endothelial-dependent dilator responses within isolated IMAs from CABG patients, and in isolated rat coronary arteries after exposure to elevated pressure in an ex vivo model of hypertension. We demonstrate that potentiation of dilation within small coronary arteries exposed to an oxidative environment (due to acute pressure elevation), are mediated by activation of NO and EDH pathways, via AMPK, in the absence of COX pathway mediators. However, in IMAs from CABG patients, TMS mediates vasodilation primarily via reduction in ROS moieties. Furthermore, we show that liposomal encapsulation of TMS improves bioavailability and enables a sustained vasodilator response in comparison to TMS solution.

A significant impairment in endothelial-dependent vasodilation was observed in IMAs harvested from CABG patients, previously suggested to be predominantly due to a reduction in NO bioavailability [10], as a result of increased superoxide generation and eNOS uncoupling, mediated in-part by NADPH oxidase [22]. Despite the reduced NO bioavailability, the long-term patency of IMA grafts has been attributed to basal release of NO and EDH [10,11], enhanced by a marked increase in COX-2 expression under inflammatory conditions. These result in increased prostaglandin vasodilators PGI_2_ and PGE_2_ [23]. Indeed, the induction of COX-2 may represent an endogenous defense mechanism against endothelial damage incurred during surgical preparation of these vessels for CABG [24]. Using ROS-sensitive fluorescent probes and immunohistochemistry, we demonstrated a high oxidative component and vessel wall remodeling within IMAs, indicating advanced disease. Other studies using rats with chronic heart failure (CHF) have previously reported increased EDH activity as a compensatory response to a reduction in NO bioavailability, which was reversed into EDH deficiency during advanced CHF, leading to enhanced vasoconstriction [25]. The release of cytochrome P450 metabolites in hypertension may contribute to this, with elevated levels of 20-HETE reported to not only induce vasoconstriction via sensitization of smooth muscle cells (SMCs) through inhibition of BKCa channels, but also attenuate the relaxant responses to ACh in phenylephrine pre-constricted arteries [26]. Moreover, the increased constrictor responses associated with endothelial dysfunction have been ascribed to the reduction in NO-mediated inhibition of the potent vasoconstrictor 20-HETE, with subsequent ROS elevation via CYP1B1-dependent NADPH oxidase activity [5,6,7,8]. Finally, the IMAs used in our study were harvested from patients with advanced disease and as discussed, the minimal dilation can be attributed to altered vasomotor balance [27], perhaps partly attributable to a defective or absent endothelial cell lining, demonstrated by our immunohistology analysis.

Acute exposure of IMAs to the potent CYP1B1 inhibitor, TMS, delivered via liposomes, led to a significant potentiation of dilator responses, confirming a key role for 20-HETEs in reducing dilator capacity in the IMAs. There is also evidence for a significant role of endothelial-independent dilation in IMAs, produced via epoxyeicosatrienoic acid activation of large conductance channels in SMCs, elicited by ACh, ultimately promoting hyperpolarization and causing relaxation [11]. This dilator mechanism is inhibited by 20-HETE, and therefore the inhibition of CYP1B1 by TMS may also potentiate the endothelium-independent dilator pathway via reduction in 20-HETE. The absence of ACh-induced dilation after dilator pathway inhibition may be because TMS blocked the cytochrome P450 pathway within the diseased environment [28].

DHE was used to specifically detect superoxide moieties, while DCF-DA enables detection of hydroxyl and peroxyl radicals within the cell. Co-incubation with TMS-loaded liposomes significantly reduced fluorescence intensities of superoxide and mitochondrial ROS and restored NO bioavailability in isolated IMAs. Together, these findings highlight the significance of ROS in attenuating vasodilation and the role of TMS-loaded liposomes in reducing these ROS moieties and potentiating NO production. To help further identify the specific vasodilator pathways potentiated by TMS-loaded liposomes, we utilized an ex vivo model of hypertension using coronary vessels from normal rats.

Using an ex vivo model, we show that acute pressure-induced impaired dilatory response was alleviated by co-incubation of isolated coronary arteries with TMS-loaded liposomes, to a degree comparable to that observed after SOD treatment. These findings highlight the antioxidant properties of TMS, reducing ROS moieties indirectly, primarily via inhibition of CYP1B1-dependent NADPH oxidase [5,6,7,8,29], which is similar to our findings in aortic vessels [18]. Additionally, TMS has previously been shown to enhance ACh-induced endothelial-dependent vasodilation and reduce vascular reactivity in spontaneously hypertensive rats and DOCA-salt induced hypertensive rats via reduction in 20-HETE and ROS generation [5,6,7,8]. Of note, the improved coronary vasodilator response to ACh was sustained over longer durations (4 h) after exposure to TMS-loaded liposomes compared to TMS solution, indicating the ability of liposomes to successfully entrap, protect and sustainably release the encapsulated TMS over a prolonged period.

Recent advances have been made into the use of liposomes as efficient drug delivery modalities due to their biocompatibility and penetration into tissues, as well as the ability to target specific tissues to reduce side effects [30,31]. Our findings add further significance to these studies and suggest that targeted PEG liposomes may represent an endothelial cell-specific drug delivery system at sites of vascular injury [32]. We propose the sustained release properties of drug-loaded liposome nanoparticles are a novel approach to enhance vasodilation in the follow-up period for CABG patients. In time, targeted delivery could overcome undesired drug effects, including pharmacokinetic complications, and allowing higher doses to reach the intended target, thus improving the cardioprotective potential of drugs in patients with cardiac ischemic events.

Our HPLC data demonstrate that while some TMS degradation was observed in TMS solution (20% degraded), it cannot be the reason for the biological difference between TMS solution and TMS-loaded liposomes. As liposomes have been widely demonstrated to cross biological membranes [31], they are expected to be a better vehicle for TMS delivery into target tissues than diffusion of TMS out of the solution or the blood stream into cells, and therefore packaging TMS into liposomes may enhance the cellular bioavailability of TMS. This may also explain the sustained dilator effects of TMS-loaded liposomes, compared to TMS solution, which may contribute to stimulation of additional signaling dilator pathways, such as hyperpolarization. The endothelial-independent responses of all arteries to SNP were maintained, suggesting that the sensitivity of SMCs is unaffected by the disease state, pressure elevation, or treatment with TMS-loaded liposomes.

We demonstrate that TMS-loaded liposomes can potentiate vasodilator responses of coronary arteries via stimulation of NO and EDH pathways following acute pressure elevation. The residual dilator responses observed in the present study following EDH inhibition may be due to a compensatory mechanism orchestrated by NO [21,33]. In order to assess whether TMS-loaded liposomes potentiate an additional dilator pathway, vessels were co-incubated with all four inhibitors in combination. Herein, the modest dilator component that remained following incubation with all four inhibitors in the presence of TMS-loaded liposomes could be due to activation of transient receptor potential vanilloid 4 (TRPV4) channels which are widely expressed in vascular endothelial cells and have been implicated in facilitating dilation in the coronary microcirculation [34]. The dilator potential of TMS-loaded liposomes was absent in denuded coronary arteries, suggesting that TMS-loaded liposomes exert their effects on the endothelium.

We further demonstrate that AMPK plays a role in mediating TMS-loaded liposome potentiation of coronary arterial dilation. We previously demonstrated that the parent molecule RV, delivered via nano-lipid carriers, activates AMPK to cause a NO-dependent dilator response [21]. AMPK plays a key role in the phosphorylation and acetylation of eNOS, leading to enhanced dilator activity via release of NO and EDH in the coronary vasculature [35], and downregulation of NADPH oxidase-derived ROS production [36]. Indeed, AMPK activation ameliorates the attenuated endothelial-dependent dilator responses of several vascular beds in hypertensive humans and experimental animal models, an improvement attributable to eNOS activation [37,38]. The residual dilation observed in the present study following AMPK inhibition, may be due to a compensatory mechanism by SIRT-1 [39,40]. Additionally, studies have established the involvement of AMPK in mediating EDH-type relaxation of resistance arteries [41,42] via activation of endothelial ATP-sensitive potassium, and large conductance calcium-activated potassium channels, resulting in SMC relaxation by hyperpolarization and lowering of intracellular Ca^2+^ [36,43]. While the current data show that acute AMPK activation by TMS-loaded liposomes improves vasodilator responses via release of NO and EDH, the implications of long-term AMPK activation, and hence therapeutic potential of TMS-loaded liposomes, are unknown. Long-term activation of AMPK in mice administrated with the chronic AMPK activator AICAR mitigated the adverse effects on eNOS by 20-HETE and protected endothelial function ex vivo [44]. Hence, deregulation of eNOS signaling and endothelial dysfunction in hypertensive disorders, associated with elevated 20-HETE levels, could be reduced via potent inhibition of CYP1B1 by TMS [5,6,7,8].

While the vascular effects of unencapsulated polyphenols are known to be mediated via activation of cell surface receptors [45], liposomal drug delivery systems are capable of being endocytosed into endothelial cells [46,47]. In the present study, co-incubation with chlorpromazine hydrochloride in the presence of TMS-loaded liposomes (mean diameter of 141 ± 0.9 nm),resulted in partial reduction in dilator responses of coronary vessels, suggesting some involvement of clathrin-mediated endocytosis in mediating uptake. Whilst a substantial amount evidence indicates clathrin-mediated endocytosis as the predominant mechanism of liposomal entry into cells, with a maximum internationalization particle size of ~200 nm [48], liposome-cell membrane fusion can also contribute to the cellular internalization of lipid-based nanocarriers and may account for the residual dilation observed in the present study [49,50]. Liposomes were prepared for intravenous administration and have previously been tested via this route in animal models using a targeted liposomal approach [47].

## 5. Conclusions

Using isolated internal mammary arteries (IMAs) harvested from CABG patients undergoing CABG surgery, and an ex vivo rat coronary artery model of acute hypertension, we provide evidence that TMS delivered via liposomes has the potential to ameliorate hypertension-associated vascular impairment, restoring attenuated vasodilation over a longer duration than TMS solution alone. Acute exposure to TMS-loaded liposomes improves vasodilator responses of the isolated IMAs primarily via inhibition of CYP1B1 and reduction in ROS moieties within the vessel wall. In isolated acute hypertensive rat coronary arteries, TMS-loaded liposomes potentiate the NO and EDH dilator pathways via AMPK. Our findings have significant implications for the future development of TMS-loaded liposomes as a therapeutic strategy to help restore vasodilator function in hypertension, with relevance for early and long-term treatment of CABG patients, via sustained and localized TMS release within IMAs.

## Figures and Tables

**Figure 1 pharmaceutics-14-02046-f001:**
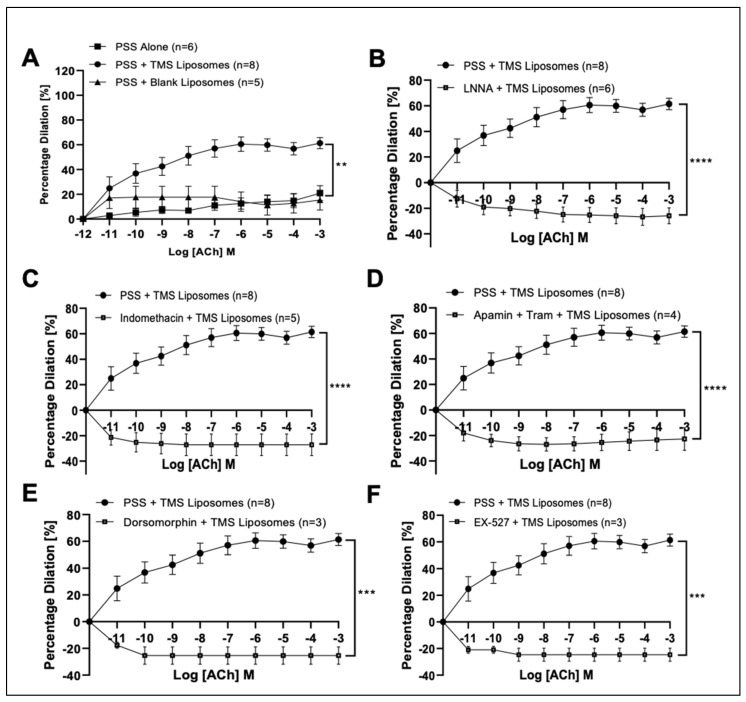
The influence of Tetramethoxystilbene (TMS)-loaded liposomes on vascular responses of isolated left internal mammary arteries (IMAs) from coronary artery bypass graft (CABG) patients ex vivo. Endothelialdependent acetylcholine (ACh; (**A**)) induced responses following coincubation with TMS-loaded liposomes. The influence of NΩ-nitro-L-arginine (LNNA; (**B**)), (Indomethacin; (**C**)), (Apamin and TRAM34; (**D**)), (5′ adenosine monophosphate-activated protein kinase (AMPK) inhibitor, Dorsomorphin; (**E**)); and (Sirtuin-1 (SIRT-1) inhibitor, EX-527; (**F**)) on ACh-induced responses following co-incubation with TMS-loaded liposomes. Physiological salt solution (PSS) Alone (*n* = 6); PSS + TMS Liposomes (*n* = 8); PSS + Blank Liposomes (*n* = 5); LNNA + TMS Liposomes (*n* = 6); Indomethacin + TMS Liposomes (*n* = 5); Apamin + Tram + TMS Liposomes (*n* = 4); Dorsomorphin + TMS Liposomes (*n* = 3); EX-527 + TMS Liposomes (*n* = 3). The area under the curve for two groups were compared using Twoway ANOVA followed by Tukey’s multiple comparisons post-test. Data are presented as mean ± SEM. ** *p* < 0.01, *** *p* < 0.001, **** *p* < 0.0001.

**Figure 2 pharmaceutics-14-02046-f002:**
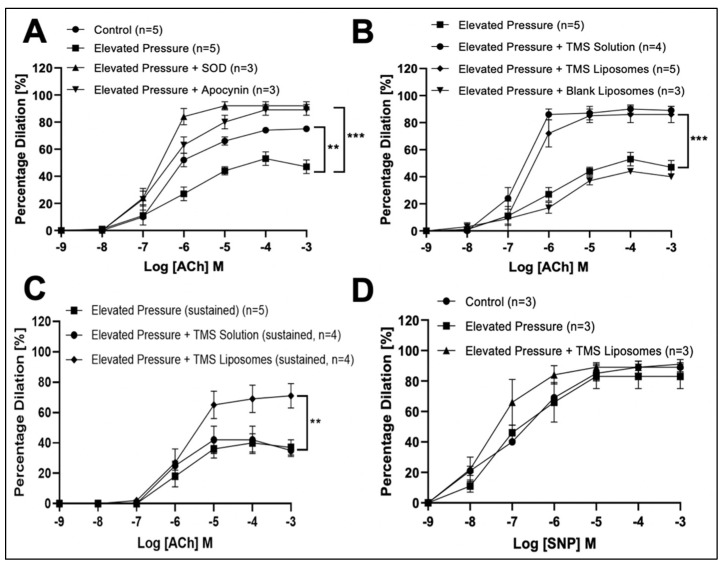
The influence of Tetramethoxystilbene (TMS)-loaded liposomes on vascular responses of isolated rat coronary arteries ex vivo. Endothelial-dependent acetylcholine (ACh)-induced responses in normotensive vessels exposed to standard pressure (control), acute pressure elevation ± superoxide dismutase (SOD) or apocynin (**A**). The influence of TMS/TMS-loaded liposomes on initial (**B**) and sustained (4 h after exposure to TMS) (**C**) on ACh-induced responses following acute pressure elevation. Endothelial-independent sodium nitroprusside (SNP)-induced responses in vessels exposed to standard pressure, acute pressure elevation ± TMS-loaded liposomes (**D**). All controls were subjected to standard pressure alone. Control (*n* = 5); Elevated Pressure (*n* = 5); Elevated Pressure + SOD (*n* = 3); Elevated Pressure + Apocynin (*n* = 3); Elevated Pressure + TMS Solution (*n* = 4); Elevated Pressure + TMS Liposomes (*n* = 5); Elevated Pressure + Blank Liposomes (*n* = 3); Elevated Pressure (sustained) (*n* = 5); Elevated Pressure + TMS Solution (sustained) (*n* = 4); Elevated Pressure + TMS Liposomes (sustained) (*n* = 4). The area under the curve for two groups were compared using Two-way ANOVA followed by Tukey’s multiple comparisons post-test. Data are presented as mean ± SEM. ** *p* < 0.01, *** *p* < 0.001.

**Figure 3 pharmaceutics-14-02046-f003:**
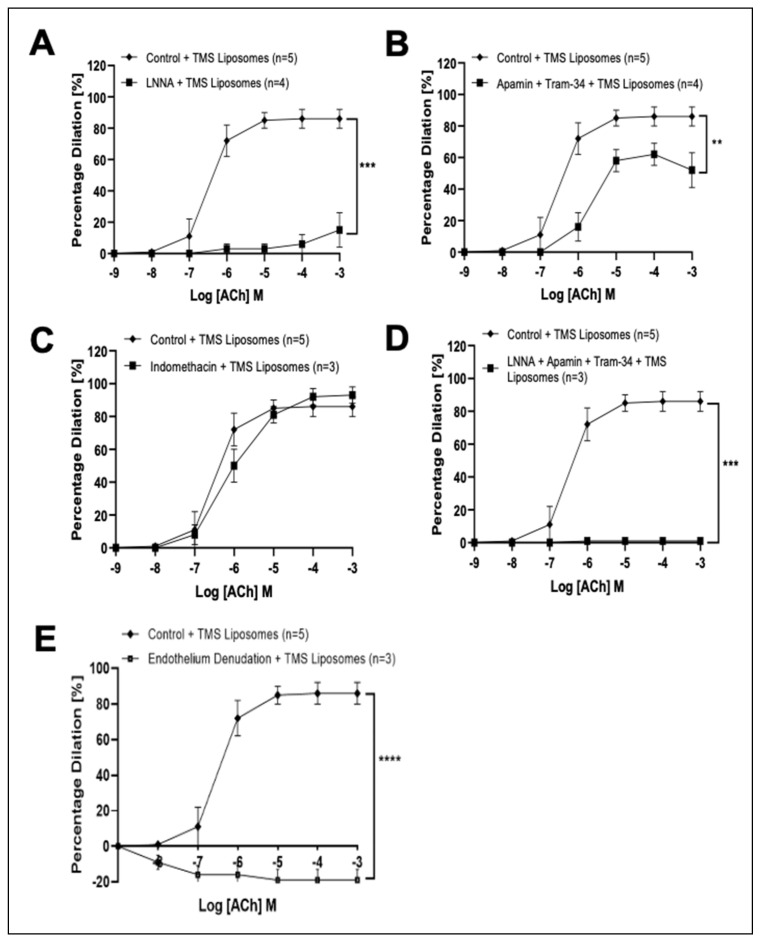
Characterization of dilator responses potentiated by Tetramethoxystilbene (TMS)-loaded liposomes following acute pressure elevation in isolated coronary arteries ex vivo. The influence of NΩ-nitro-L-arginine (LNNA; (**A**)), (Apamin and TRAM-34; (**B**)), (Indomethacin; (**C**)), (L-NNA, Apamin and TRAM-34; (**D**)) and (after endothelial denudation; (**E**)) on endothelium-dependent acetylcholine (ACh) responses following acute pressure elevation. Control + TMS Liposomes (*n* = 5); LNNA + TMS Liposomes (*n* = 4); Apamin + Tram-34 + TMS Liposomes (*n* = 4); Indomethacin + TMS Liposomes (*n* = 3); LNNA + Apamin + Tram-34 + TMS Liposomes (*n* = 3); Endothelium Denudation + TMS Liposomes (*n* = 3). The area under the curve for two groups were compared using Two-way ANOVA followed by Tukey’s multiple comparisons post-test. Data are presented as mean ± SEM. ** *p* < 0.01, *** *p* < 0.001, **** *p* < 0.0001.

**Figure 4 pharmaceutics-14-02046-f004:**
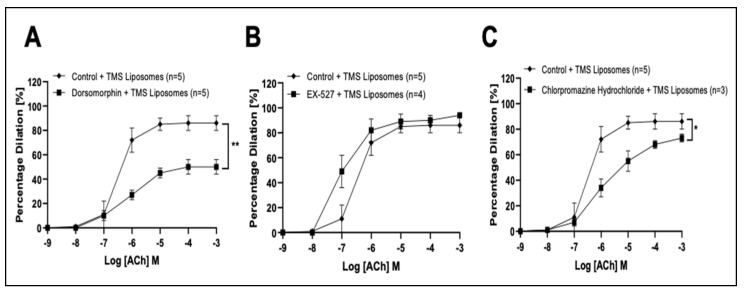
Characterization of the molecular mechanisms stimulated by Tetramethoxystilbene (TMS)-loaded liposomes following acute pressure elevation in isolated coronary arteries ex vivo. The influence of (5′ adenosine monophosphate-activated protein kinase (AMPK) inhibitor, Dorsomorphin; (**A**)), (Sirtuin-1 (SIRT-1) inhibitor, EX-527; (**B**)); and Chlorpromazine hydrochloride; (**C**)) on acetylcholine (ACh)-induced dilation in the presence of tetramethoxystilbene (TMS)-loaded liposomes following acute pressure elevation. Control + TMS Liposomes (*n* = 5); Dorsomorphin + TMS Liposomes (*n* = 5); EX-527 + TMS Liposomes (*n* = 4); Chlorpromazine Hydrochloride + TMS Liposomes (*n* = 3). The area under the curve for two groups were compared using Two-way ANOVA followed by Tukey’s multiple comparisons post-test. Data are presented as mean ± SEM. * *p* < 0.05, ** *p* < 0.01.

**Figure 5 pharmaceutics-14-02046-f005:**
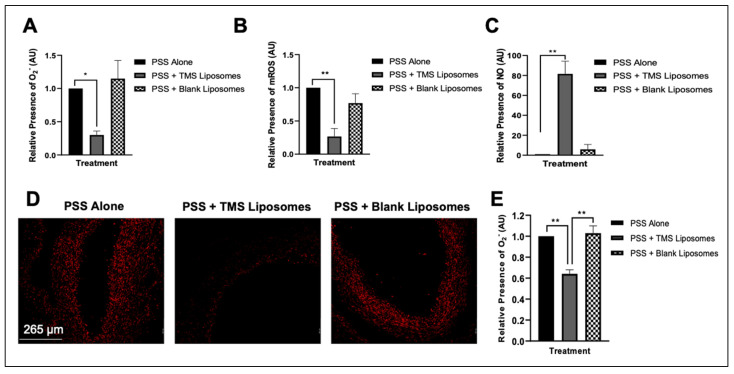
The corrective influence of Tetramethoxystilbene (TMS)-loaded liposomes on levels of reactive oxygen species (ROS) moieties and nitric oxide (NO) bioavailability within isolated left internal mammary arteries (IMAs) from coronary artery bypass graft (CABG) patients. TMSloaded liposomes reduce relative fluorescence of superoxide anion (O^2−^) (Dihydroethidium fluorescence, DHE; (**A**)) and mitochondrial ROS (MitoSOX; (**B**)) and increases intracellular NO as measured by Diamino fluorescein-FM diacetate (DAF-FM DA; (**C**)) in isolated IMAs. Fluorescence data are expressed as mean fluorescence intensity (percentage of control). *n* = 4. Representative fluorescence micrographs (**D**) and semiquantitative analysis (**E**) of DHE signal intensities in IMA sections. In the absence of TMS-loaded liposomes, strong fluorescent DHE signals were observed, while vessels incubated with TMS-loaded liposomes exhibited a weak DHE signal. Images are representative for 10 sections per vessel: *n* = 3 vessels per group. Values for semiquantitative analysis of fluorescence signal intensities were normalized to IMA controls (physiological salt solution (PSS) alone). Magnification ×10. Scale bar = 265 µm. Oneway ANOVA followed by a Dunnett’s multiple comparisons test. * *p* < 0.05, ** *p* < 0.01.

**Figure 6 pharmaceutics-14-02046-f006:**
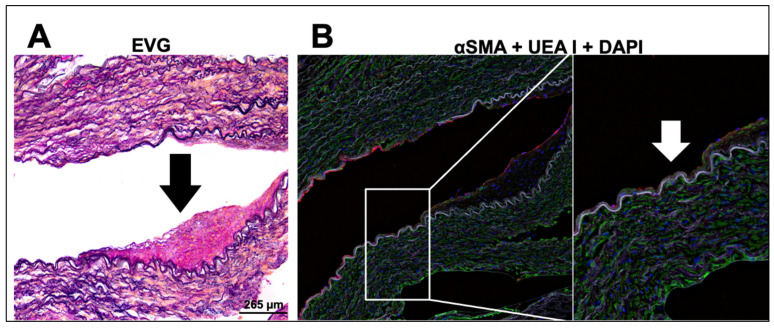
Photomicrographs of diseased human internal mammary artery (IMA) from a coronary artery bypass graft (CABG) patient. Representative serial sections of IMA stained with (**A**) Elastic Van Gieson (EVG), (**B**) alpha smooth muscle actin (α-sma) (green fluorescence), Ulex Europaeus Agglutinin I (UEA I) (red fluorescence) and DAPI (detection of nuclei) (blue fluorescence). Increased matrix deposition of elastin on the elastic lamina was indicated by a black arrow. Magnification ×10. Scale bar = 265 µm. The representative absence of an intact endothelial cell lining, as shown by the lack of red fluorescence, is indicated by a white arrow (insert; magnification ×20). These are representative images of similar observations in five patients with 10 sections from each patient.

## Data Availability

The data presented in this study are available on request from the corresponding authors.

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
