# Peer review of "Internal Mammary Arteries as a Model to Demonstrate Restoration of the Impaired Vasodilation in Hypertension, Using Liposomal Delivery of the CYP1B1 Inhibitor, 2,3′,4,5′-Tetramethoxystilbene"

_pharmaceutics, 2022, doi:10.3390/pharmaceutics14102046_

Round 1

Reviewer 1 Report

In this manuscript, Azzawi and coworkers studied the liposomal delivery of 2,3’,4,5’-tetramethoxystilbene (TMS) to determine the inhibition of P450 enzyme CYP1B1 and subsequent vasodilation of IMAs from CABG patients with PEGylated liposomes. The proposed mechanism was carefully evaluated in an ex vivo model of hypertension. Overall, the manuscript is well-written, and the experiments are well designed. The reviewer thinks this work is an important finding in the field and suits the scope of Pharmaceutics. However, below are some minor suggestions the authors can think about to improve the manuscript further:

1.    In Figures 1B-1F and Figure 3F, some of the curves overlap with the x-axis, making them hard to read. The authors should work on the figures to avoid this issue. 

2.    In section 2.1, the authors mentioned the encapsulation efficiency and cited ref 17. However, it seems no such info is included in ref 17, and they were dealing with different content. Also, 50% is an exceptional high encapsulation efficiency (%EE) for passively encapsulated hydrophobic contents. Since %EE can be different from system to system, it would be better if the authors could measure this value again instead of just citing the previous literature. 

Reviewer 2 Report

See attachment. 

Reviewer 3 Report

1. 2.1. Synthesis of TMS-Loaded Liposomes: Please rewrite the process of manufacturing liposomes clearly, so can be possible to reproduce by other researchers. What was the size of blank and loaded liposomes, mention in the discussion?

2. Were liposomes prepared for injectable route or oral, please mention in the manuscript?

3. 2.3.1. Human Internal Mammary Artery (IMA) Functional Studies and 2.3.2. Rat Coronary Artery Functional Studies, can this experiment possible to perform in others animal artery, if yes, then which animal can use? What is the correlation between human mammary artery and rat coronary artery?
